# Simulation-Based Gastrointestinal Endoscopy Sedations: A Novel Validation to Multidrug Pharmacodynamic Modeling

**DOI:** 10.3390/pharmaceutics14102056

**Published:** 2022-09-27

**Authors:** Jing-Yang Liou, Hsin-Yi Wang, I-Ting Kuo, Wen-Kuei Chang, Chien-Kun Ting

**Affiliations:** 1Department of Anesthesiology, Taipei Veterans General Hospital, No. 201, Sec. 2, Shipai Rd., Beitou District, Taipei City 112201, Taiwan; 2School of Medicine, National Yang-Ming Chiao Tung University, Taipei City 112304, Taiwan; 3Department of Biomedical Engineering, National Yang-Ming Chiao Tung University, Taipei City 112304, Taiwan; 4Department of Biomedical Sciences and Engineering, National Central University, Taoyuan City 320317, Taiwan; 5Institute of Emergency and Critical Care Medicine, National Yang Ming Chiao Tung University, Taipei 112304, Taiwan

**Keywords:** computer model simulation, conscious sedation, deep sedation, moderate sedation, pharmacodynamics

## Abstract

Pharmacodynamic models have described the interactions between anesthetics. Applying the models to clinical practice is still problematic due to inherent limitations: 1. modeling conditions are different from practice. 2. One model can only describe one endpoint. To tackle these, we propose a new method of model validation for recovery and intraprocedural sedation adequacy with a three-drug pharmacodynamic model using six published clinical studies that contain midazolam, opioid, and propofol. Mean drug dose, intraprocedural sedation level, procedure, and recovery time are extracted from each study. Simulated drug regimens are designed to best approximate study conditions. A published deep sedation model is used for simulation. Model-predicted recovery time and intraprocedural sedation scores are compared with the original clinical study outcomes. The model successfully predicted recovery times in eight out of nine regimens. Lower doses of midazolam are associated with faster recovery. Model prediction of intraprocedural sedation level was compatible with the clinical studies in five out of seven regimens. The three-drug pharmacodynamic model describes the course of gastrointestinal endoscopy sedations from clinical studies well. Model predictions are consistent with the results from clinical studies. The approach implies that large scale validation can be performed repeatedly.

## 1. Background

Drugs for sedation are characterized by rapid onset and offset, and narrow margins of safety. Significant drug interactions are expected when multiple drugs are given simultaneously. Interactions are divided into additivity, synergism, or infra-additivity [1]. Two-drug interactions were investigated extensively in anesthesia, confirming synergism in most drug combinations [2,3,4,5,6,7,8]. Three-drug models are more complex [9,10], but more suitable for clinical scenarios. These pharmacodynamic response surface models give insights into better dosing strategies and make multidrug simulations possible [11], but there are inherent problems that preclude their clinical application: model generalization and endpoint limitations.

Generalization is required for clinical use. Pharmacodynamic models are trained and validated by volunteers or patients from the same institution. Patient characteristics are similar in both training and validation groups. Drug regimens are fixed to follow predefined study protocols. This approach also raises uncertainties in model performance outside the original study conditions [12].

A model is linked with a single predefined categorical endpoint. For instance, a deep sedation model for loss of response (LOR), defined by the observer’s assessment of alertness/sedation score (OAA/S) = 1 [13] (Table 1), is theoretically unfit for light sedation. Clinically relevant endpoints would need frequent parameter conversions. This greatly hindered the model’s clinical use.

In an attempt to overcome the aforementioned inherent issues in modeling, we propose a novel approach for validating the performance of a single multidrug pharmacodynamic model using published literature. Specifically, we aim to simulate sedation protocols with combinations of midazolam, propofol, and an opioid for gastrointestinal endoscopies from other clinical studies to predict both times of recovery using only a set of parameters. Intraprocedural sedation adequacy is assessed. Drug regimens, sedation scores, procedure time, and time to recovery are extracted from each clinical study. Agreement between model prediction and published study results are assessed.

## 2. Methods

### 2.1. Search Strategy and Article Identification

We conducted a literature search in PUBMED from 2000 to 2022 for three-drug sedation regimens. Broad range search with keywords propofol, sedat *, endoscop *, and colonoscop * were used and limitation assigned to study drugs. The eligible criteria for inclusion were:(1)Contained at least one three-drug combinational regimen. If an identified study included both three- and two-drug regimens, both regimens are qualified for simulation.(2)The three drugs must be intravenous midazolam, propofol, and an opioid (alfentanil, fentanyl, remifentanil, or morphine).(3)Studies were performed for gastrointestinal endoscopy.(4)Studies must report procedure and recovery time.(5)English literature.

Articles were excluded if patients were morbidly obese, cirrhotic, octogenarians, pediatrics, or had a history of congestive heart failure. Duplicates were removed and abstracts/titles of all articles were screened by J.Y.L. and H.Y.W. using inclusion criterion. This step was followed by reading the remaining full-text articles out of which relevant articles. Two reviewers (J.Y.L. and H.Y.W.) independently assessed the full texts of potentially relevant studies using the inclusion criterion. Only research articles were considered. In this protocol, it was ideal to include studies with different sedation schemes to demonstrate model credibility.

### 2.2. Simulation Setup

To illustrate the clinical study sedation course, the study-reported mean procedure times (T_p_) and recovery times (T_rs_) in each regimen are used to best approximate the original study conditions. Simulations start with 3 min induction time [14,15,16,17]. The observed recovery time (T_ro_) is defined by Equation (1). T_ro_ range is calculated by adding and subtracting T_rs_ standard deviations (SD) (Equation (1)).
(1)$$\begin{array}{c} T_{\mathrm{ro}}=\;\mathrm{Induction}\;\mathrm{time}\;+{\;T}_{p}+{\;T}_{\mathrm{rs}}\\ T_{\mathrm{ro}}\;\mathrm{range}=\mathrm{Induction}\;\mathrm{time}\;+{\;T}_{p}+{\;T}_{\mathrm{rs}}\;\pm {\;T}_{\mathrm{rs}}\;\mathrm{SD} \end{array}$$

The 5% probability of MOAA/S (Modified OAA/S, Table 1) < 2 is used as a cutoff to identify model predicted recovery from sedation for the simulation. The time from the start to the predicted recovery is designated as model predicted recovery time (T_rm_).

The total doses are determined by the study-reported mean doses. Dosing follows the strategies:(1)Midazolam typically lasts 30 to 60 min [18]. The dose is not divided and given at induction.(2)Fentanyl reaches peak effect-site concentration (Ce) at 3.6 min and more than 50% Ce remains at 30 min [19]. The dose is not divided and given at induction.(3)Alfentanil reaches peak Ce at 1.4 min, with a rapid decline over the first ten minutes after a bolus [19]. It is divided into equivalent doses at 10 min intervals and avoided 10 min before the procedure concludes.(4)Propofol’s time to peak effect (TTPE) is 1.6–1.7 min [20]. The total dose is divided into boluses at 2 to 3 min intervals or greater if the total dose is small. It is avoided in the final two minutes.

### 2.3. Pharmacologic Models

The pharmacokinetic and pharmacodynamic models have been developed and validated through comprehensive study designs. Pharmacokinetic models are selected based on availability and clinical practice. Drug concentration effects are calculated with the models. Opioids are converted to alfentanil equivalents (Appendix A) [1,9,17,21,22,23,24,25,26,27]. All drug concentrations are expressed as Ce.

### 2.4. Outcome Assessment

The primary aim is to compare the T_rm_ with T_ro_ (Equation (1)). Two approaches are used for validation. First, the model is considered accurate if T_rm_ falls within the T_ro_ range. Second, we calculate the absolute prediction error (T_rm_ − T_ro_). The model is considered accurate if the absolute prediction error falls within half of the average T_rs_ margin. An absolute prediction error percentage is presented to illustrate the normalized degree of error and is calculated by (T_rm_ − T_ro_)/T_ro_. The two approaches represent individualized and population pooled measurements. Validation also examines if a LOR model (MOAA/S < 2) model is fit to predict patient recovery (MOAA/S = 4 or 5) without the need for additional modeling.

Secondly, we examine how well our model predicts intraprocedural sedation. Model LOR (MOAA/S < 2) probability greater than 50% indicates deep sedation, while probability less than 50% indicates moderate sedation. Intraprocedural proportion of deep and moderate sedation is calculated. Model is accurate if the dominating predicted sedation depth agrees with the clinical studies.

## 3. Results

### 3.1. Study Eligibility Search Results

PUBMED search identified a total of 627 records. The removal of 121 duplicates left 506 articles. Out of these, 483 articles were excluded based on title and abstract screening. From the screening, 23 full-text articles were then assessed for eligibility, and 17 studies were excluded. Among the excluded studies, most of them were due to violation of inclusion criteria item 2. This study selection resulted in a total of six eligible full articles, which were all included in our study [16,17,21,28,29,30]. All three- and two-drug regimens were included, giving a total of nine regimens. Study summaries were listed in Table 2 and dosing strategies in Table 3.

The sedation score used by Chan was used by Bill et al., which closely resembled the OAA/S score in a reversed fashion. Recovery time deviation referred to the standard deviation given by the individual studies. Chan et al. did not report a standard deviation, and thus a range of -20 min was adopted, details were described in Methods.

EGD: esophagogastroduodenoscopy; ERCP: endoscopic retrograde cholangiopancreatography); (M)OAA/S: (modified) observer’s assessment of alertness/sedation scale; T_p_: procedure time (mean); T_rs_: study reported recovery time (mean); SD: standard deviation.

Simulation starts at 0 min, with an induction period of 3 min. Procedure time is simulated according to the mean procedure time given by individual clinical studies. Simulation ends at 60 min.

Simulation patient: female, height 170 cm, weight 65 kg, ASA 2.

### 3.2. Recovery and Time Definition Modification in the Identified Studies

Chan assessed sedation with a scale from Bill et al. [31], which was very similar to OAA/S in a reversed fashion. A score of 1 would correspond to MOAA/S = 5. In Regimen 5, T_rs_ and its SD were not reported. Instead, the percentage of patients reaching certain sedation scores was available at 0, 5, 10, and 30 min. The time when 90% of the individuals reached a score = 5 was used as T_rs_, which is at 30 min (45 out of 50 patients). Only 41 of 50 patients reached a score = 5 at 10 min. Intuitively, the four patients must have reached a score of 5 between 10 and 30 min. Thus, the 20 min time gap was used as our one-sided T_rs_. The same principle was applied to regimen 6, where 47 out of 60, and 57 out of 60 patients reached a score = 5 at 10 and 30 min, respectively. T_rs_ was 30 min and T_rs_ SD was 20 min.

### 3.3. Recovery Profile of the Clinical Studies and Model Comparison

T_ro_ and T_ro_ ranges were listed in Table 4. Figure 1 was a graphical presentation of the time course. T_rm_ (red cross in Figure 1) were within the T_ro_ range in all simulations except regimen 3. In regimen 3, our model predicted recovery 5.58 min earlier than the T_ro_ range. Adjustments were examined (not shown) where the 50 mg of propofol were dispersed in different proportions for regimen 3 but T_rm_ remained outside the T_ro_ range.

The average pooled T_rs_ was 14.9 min. We used 7.4 min as the margin of accuracy. The absolute prediction errors were listed in Table 4 where all, except regimen 3, were accurate. Accurate predictions were within 18% deviation from T_ro_.

Higher midazolam doses delay recovery more pronounced than propofol or opioids. T_rm_ (red cross in Figure 1) occurred on average 1.56 min after the procedure ended in regimens 2, 3, 7, and 8. These regimens had lower midazolam doses that ranged from 0 to 1.1 mg. T_rs_ were also shorter in regimens 2, 3, 7, and 8. Regimen 2 has the highest propofol dose but has one of the faster recoveries (T_rs_ = 7.3 min). In contrast, T_rm_ occurred on average 21.62 min after the procedure ended for regimens 1, 4, 5, 6, and 9. Regimen 9 has a low opioid dose, but recovery was late (T_rs_ = 18.37 min). Interestingly, midazolam and fentanyl doses were almost identical in regimens 3, 7, and 8. Regimens 7 and 8 consumed higher doses of propofol, but T_rs_ was shorter. The total opioid and propofol doses did not correlate with the time to recovery. Clinical observations were consistent with our model simulation.

Only two drugs were used in regimens 2 and 4. Regimen 2 was characterized by rapid recovery (T_rs_ = 7.3 min). Regimen 4 had a long recovery (T_rs_ = 23 min). T_rm_ paralleled T_rs_ and was accurate in both settings.

Time course of the procedures and their recoveries are illustrated. Green boxes represent the procedure periods, and the purple boxes are the T_ro_ range, or the range for accurate recovery. Yellow bars indicate the reported mean recovery time from the studies. Blue lines describe the probability of MOAA/S < 2 (deep sedation) during the entire course. Red crosses (×) stand for the model predicted recovery at MOAA/S = 5%. The model predicted recovery that is consistent with clinical studies in all but regimen 3.

T_ro_: Observed Recovery Time reported by the clinical studies; MOAA/S: Modified Observer’s Assessment of Alertness/Sedation scale.

T_ro_ range was calculated by adding and subtracting from the recovery time standard deviation given by the individual clinical studies. A one-way 20 min range was used for Chan et al.; details are described in Methods. Intraprocedural deep sedation ratio was determined by the proportion of deep sedation (>50% probability of MOAA/S < 2) during the procedure time from simulation.

T_rm_: model predicted T_r_; T_ro_: observed recovery time.

The time course of the procedures and their recoveries are illustrated. Green boxes represent the procedure periods, and the purple boxes are the T_ro_ range or the range for accurate recovery. Yellow bars indicate the reported mean recovery time from the studies. Blue lines describe the probability of MOAA/S < 2 (deep sedation) during the entire course. Red crosses (×) stand for the model predicted recovery at MOAA/S = 5%. The model predicted recovery is consistent with clinical studies in all but regimen 3.

T_ro_: observed recovery time reported by the clinical studies; MOAA/S: modified observer’s assessment of alertness/sedation scale.

### 3.4. Model Intraprocedural Sedation Performance

Intraprocedural sedation scores were compared with model predictions (Table 4). Regimens 3, 4, 8, and 9 targeted moderate sedation. Our model showed 100%, 39%, 74%, and 0% of the procedure time were within the moderate sedation range (LOR probability < 50%) for regimens 3, 4, 8, and 9, respectively. Regimens 5, 6, and 7 targeted deep sedation, whereas our model showed 94%, 100%, and 83% of the procedure time in the deep sedation range, consistent with study findings. Regimens 1 and 2 did not report sedation score. To summarize, our model performed well for regimens 3, 5, 6, 7, and 8.

## 4. Discussion

We demonstrated a pragmatic and novel validation method of a single NLMAZ three-drug pharmacodynamic model using published clinical studies through simulation. The model focused on assessing drug interaction predictions of two or three drugs. Model predictions of recovery and intraprocedural sedation scores correlated well with clinical studies. Prediction of multiple endpoints with a single model was feasible. This approach allowed repeated model performance checks without the need of new patient or volunteer recruitments.

A well-developed multidrug model would be fit for patient response predictions, simulation training sessions, outcome exploration with different regimens [11], preanesthetic planning and implementation in computer-assisted sedation [32], particularly when handling multidrug regimens. Response prediction had been described. Simulation trainings served as a safe learning tool for trainees. Drug delivery was a component of anesthesia simulation [33,34,35], but most simulations dealt with pharmacokinetics (drug concentrations) or single drugs [33]. It would greatly improve simulations to include a three-drug model that complies with balanced propofol sedation (BPS) [36], as demonstrated in the article.

There were few three-drug regimens published for gastrointestinal endoscopy sedation [37]. Interactions of anesthetic drugs are primarily synergistic between two classes of drugs [8]. With three drugs, the interplay was more complex. Synergism was reported between the pairwise and triple combinations of midazolam, alfentanil, and propofol with traditional isobolographic methods [38]. Additional synergism was observed beyond binary drug pairs. It was later reconfirmed by another study using response surface methodology [9]. Response surface models allowed simulations to be performed [11,13,33]. Earlier simulation studies did not compare results with clinical studies [11].

Two- and three-drug regimens were included. Both were common strategies in practice. There was also a considerable difference between drug doses and types in the selected regimens. The feature did not affect our study aims. Variability was welcomed with our method. We reasoned that the model’s generalizability would be assured by the inclusion of a wider variety of regimens.

Recovery predictions were not accurate for regimen 3. Our model prediction was early and outside the T_ro_ range. Regimens 3 and 4 were from the same study [21]. It was partially explained by endpoint definitions. Recovery was defined by MOAA/S = 5 and Aldrete score = 10 in the original study. The Aldrete score [39] contained several unmodeled parameters and was used as an index for discharging a patient, with 10 referring to complete recovery to the patient’s baseline status. The difference between regimens 3 and 4 was the administration of propofol in regimen 3. There are reports of delayed recovery of Aldrete score after anesthesia with propofol [40,41], possibly from the delayed return of muscle activity. The components of the Aldrete score were not a part of the model and the delayed effects could not be foreseen by the model.

Intraprocedural sedation scores were accurate in five out of seven regimens. The model failed to reflect intraprocedural sedation scores in regimens 4 and 9. Stable MOAA/S = 3 without variations was reported for regimens 3 and 4 throughout the procedure [21]. Sedation score changes were usually observed as a result of drug concentration or stimuli fluctuations and interindividual variations [42]. A consistent state of sedation was hard to achieve, particularly with bolus medications in clinical settings. Twice as many patients in the regimen 4 group were deeply sedated than regimen 3 (original study arms). The trend was described by our model, where the ratio of deep sedation was higher in regimen 4.

Regimen 9 [16] targeted moderate sedation but the model predicted deep sedation. The study definition of moderate sedation included purposeful hand movement, which also qualified as deep sedation [21]. Interestingly, regimens 7 and 9 had very similar doses of fentanyl (52.5 μg vs. 50 μg), propofol (159 mg vs. 145.64 mg), and procedure time (25 vs. 28 min). Despite a higher dose of midazolam than regimen 7, regimen 9 was reported to be in moderate sedation. Both regimens consumed high doses of propofol. The literature [43] has reported deep sedation with similar amounts of propofol. Therefore, we reasoned that the study reported sedation scores that did not agree with our predictions might have risen from interrater variations. Other demographic variables were considered but they were unable to explain the difference between observation and prediction.

Propofol monosedation represented the mainstay strategy for gastrointestinal endoscopy sedation [44]. Many single propofol pharmacodynamic models were available [45] and extensively reviewed for their drug effects [20]. Literature describing multidrug pharmacodynamic studies for gastrointestinal endoscopy sedation was scarce [9,11], and even less for validation studies. Results from this study improved our understanding of the pharmacodynamic model’s clinical applications.

The accuracy definition described in the article originated from two standpoints. The model was tested for its performance with individualized SD targets and population pooled average recovery time. The halved average recovery time margin of 7.4 min was designed as a more stringent cutoff than the original average recovery time 14.9 min. Both strategies confirmed equal accuracy of the model in clinical scenario simulations. Recovery prediction was successful in eight out of nine regimens and five out of seven for sedation scores. The overall accuracy was 81%. Traditionally, the models have been validated with single events such as recovery or instrumentation for their accuracy. They have reported 79~81% accuracy with response surface models [9,13]. Our model described both recovery events and sedation scores along the course of sedation and showed good agreement between model predictions and clinical study results.

The study had several limitations that merited discussion. Single drug regimens were purposefully excluded. We intended to validate the NLMAZ to confirm its performance on drug interactions and single drugs were outside the scope of the study.

The dosing strategy was designed to best approximate practice from the given average doses, but exact mirroring was not possible. Our design is based on pharmacologic properties. Unconventional dosing such as midazolam administration near the end of the procedure was avoided. We reasoned that our approach reflected clinical practice.

Model-defined endpoints were arbitrary. Deep sedation was defined as model LOR probability above 50%, moderate sedation as probability less than 50%, and recovery was defined as less than 5%. The 5%, 50%, and 95% isoboles were common landmarks for response surface analyses [13,46]. This group of cutoff values was reasonable and described the procedure sedation course well.

The study was not aimed at individual patient predictions but rather presented as an accurate tool for population pharmacodynamic description of gastrointestinal endoscopy procedures. Significant pharmacodynamic interindividual variations should be anticipated and that flexible user interpretation was required to accurately fit the model in individual patients.

## 5. Conclusions

We successfully presented a novel method for validating a pharmacodynamic model using published clinical studies. We also demonstrated that a single model can be used to describe the course, both sedation adequacy and recovery, of procedural sedation. Our approach reduced the need for strict modeling–validation coupling during model development. Large scale validation can be performed repeatedly. It potentially simplified pharmacodynamic research and reduced patient or volunteer exposure to anesthetic drugs. The model development process is greatly shortened and made the models available for clinical use at a quicker pace. We believe the models would bring meaningful clinical guidance and educational applications in sedation management.

## Figures and Tables

**Figure 1 pharmaceutics-14-02056-f001:**
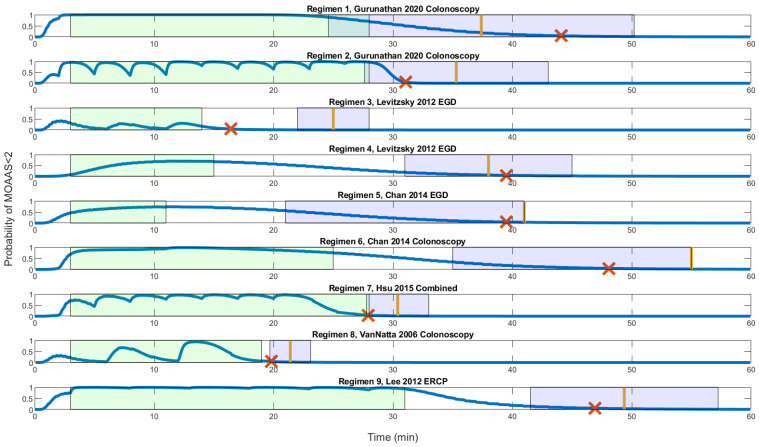
Comparison between model simulation and clinical study observations.

**Table 1 pharmaceutics-14-02056-t001:** Modified observer’s assessment of alertness/sedation (OAA/S) scale.

Observation	Score
Responds readily to name spoken in normal tone	5
Lethargic response to name spoken in normal tone	4
Responds only after name is called loudly and/or repeatedly	3
Responds only after mild prodding or shaking	2
Does not respond to mild prodding or shaking	1
Does not respond to trapezius squeeze (pain)	0

**Table 2 pharmaceutics-14-02056-t002:** Summary of the clinical studies.

Regimen	Sedation Target	Recovery Evaluation	Recovery Target	Procedure	T_p_ (min)	T_rs_ (min)	T_rs_ SD (min)
1 [21]	Not reported	Eye opening	Eye opening	Colonoscopy	25	9.4	12.8
2 [21]	Not reported	Eye opening	Eye opening	Colonoscopy	25	7.3	7.7
3 [22]	Moderate	MOAA/S	5	EGD	11	11	3
4 [22]	Moderate	MOAA/S	5	EGD	12	23	7
5 [23]	Deep	Reverse OOA/S	1	EGD	8	30	–20
6 [23]	Deep	Reverse OOA/S	1	Colonoscopy	22	30	–20
7 [24]	Deep	MOAA/S	5	Combined EGD and colonoscopy	25	2.4	2.6
8 [17]	Moderate	MOAA/S	Talking	Colonoscopy	16	2.4	1.7
9 [16]	Moderate	Modified Aldrete score	10	ERCP	28	18.37	7.86

**Table 3 pharmaceutics-14-02056-t003:** Regimen dosing summary.

Regimen	Study Protocols	Simulation Setup
1 [21]	Midazolam: 0.04 mg/kg	Midazolam: 2.5 mg
	Fentanyl: 77.5 μg	Fentanyl: 78 μg
	Propofol: 276 mg	Propofol: 30 mg at 0, 2, 4, 6, 8, 10, 12, 14, 16 min
2 [21]	Fentanyl: 66.9 μg	Fentanyl: 67 μg
	Propofol: 329 mg	Propofol: 40 mg at 0, 11, 23 min; 30 mg at 2, 5, 8, 14, 17, 20, 26 min
3 [22]	Midazolam: 1 mg	Midazolam: 1 mg
	Fentanyl: 50 μg	Fentanyl: 50 μg
	Propofol: 10 mg at induction, 5–10 mg every 30 s after assessment, total 50 mg	Propofol: 30 mg at 0 min; 10 mg at 6, 11 min
4 [22]	Midazolam: 3 mg	Midazolam: 3 mg
	Fentanyl: 100 μg	Fentanyl: 100 μg
5 [23]	Midazolam: 3 mg	Midazolam: 3 mg
	Alfentanil: 600 μg	Alfentanil: 600 μg
	Propofol: 10 mg	Propofol:10 mg at 0 min
6 [23]	Midazolam: 3.8 mg	Midazolam: 3.8 mg
	Alfentanil: 800 μg	Alfentanil: 400 μg at 0, 10 min
	Propofol: 23 mg	Propofol: 10 mg at 2, 11 min
7 [24]	Midazolam: 1.1 mg	Midazolam: 1.1 mg
	Fentanyl: 52.5 μg	Fentanyl: 52.5 μg
	Propofol: 159 mg	Propofol: 20 mg at 0, 2, 5, 8, 11, 14, 17, 20 min
8 [17]	Midazolam: 1 mg	Midazolam: 1 mg
	Fentanyl 50 μg	Fentanyl: 50 μg
	Propofol: 82.5 mg	Propofol: 30 mg at 0, 12 min; 20 mg at 6 min
9 [16]	Midazolam: 0.05 mg/kg	Midazolam: 3.25 mg
	Fentanyl: 50 μg	Fentanyl: 50 μg
	Propofol: 145.64 mg	Propofol: 20 mg at 0, 8, 13, 18, 23 min; 30 mg at 3 min; 15 mg at 28 min

**Table 4 pharmaceutics-14-02056-t004:** Simulation and clinical study recovery times.

Regimen	T_rm_ (min)	T_ro_ (min)	T_ro_ Range (min)	Ratio of Intraprocedural Deep Sedation
1 [21]	44.1	37.4	24.6–50.2	-
2 [21]	31.1	35.3	27.6–43.0	-
3 [22]	16.4	25.0	22.0–28.0	0.00
4 [22]	39.5	38.0	31.0–45.0	0.71
5 [23]	39.5	41.0	21.0–41.0	0.94
6 [23]	48.1	55.0	35.0–55.0	1.00
7 [24]	27.9	30.4	27.8–33.0	0.83
8 [17]	19.8	21.4	19.7–23.1	0.26
9 [16]	46.9	49.4	41.5–57.2	1.00

## Data Availability

Not applicable.

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
