# Peer review of "Simulation-Based Gastrointestinal Endoscopy Sedations: A Novel Validation to Multidrug Pharmacodynamic Modeling"

_pharmaceutics, 2022, doi:10.3390/pharmaceutics14102056_

Round 1

Reviewer 1 Report

The manuscript describes a novel approach for validation of the previously proposed by White et al. (Curr Drug Metab., 2003) pharmacodynamic model for three drugs given simultaneously to patients undergoing gastrointestinal endoscopies. The data for analysis were taken from literature. The study results may be interesting for clinicians, however, the manuscript is very difficult to follow and it is not possible to repeat the procedures used due to the lack of detailed information on the methodology.

First of all, the model used was not described in detail. It is not clear how the pharmacokinetic or pharmacodynamic interactions were incorporated in the model.

In Supplementary materials can be found that  “There are a total of 23 parameters for the MOAA/S < 2 model used for the simulation”. These parameters were not listed and their values used for simulations were not provided. From eq. 2 the number of parameters seems much lower.

In the sentence: “C50 resides in U50and stands for the concentration of a drug to reach half maximal effect” it is not clear whether C50 is plasma or effect site concentration.

C50mid=51.215, C50alf=627.933 and C50prop=2.581” – units are missing. They were called primary parameters, whereas they are not present in eq. 2 as model parameters.

Line 113 – “Pharmacokinetic models are selected based on clinical relevance.” Usually they are selected based on the goodness of fit criteria. The PK data and models should be presented and model selection justified based on the standard criteria.

Line 114 – “Drug concentrations effects are calculated with the models”. How were effects calculated with PK models?

The Authors have not presented how the study results may be used in clinical practice.

Author Response

(the responses are best read by word processors)

Reviewer 1

  Thank you for the valuable comments. Pharmacologic studies involve modeling and validation, which are usually performed in the same study. This is true for both PK and PD. In this article, we focused on the validation part of PD, and published PK and PD models were used in the process. The PD model was developed for anesthesia use by our group in 2018 (Br J Anaesth 2018; 120(6): 1209-1218).

  For a model to be accurate in clinical practice, it must fit different types of patients and procedures to widen its applications. Traditional validation processes used volunteers/patients that has very similar demographics as the modeling counterpart. This is primary due to single hospital practice standards and the need to follow pre-defined study protocols. The PD model’s accuracy outside this group of volunteers/patients is unknown. We believe there’s a need for new validation method for this purpose. We tackled this problem in this study by including different published endoscopy studies. There were substantial procedural and patient variabilities in the article, which may appear unconventional for PKPD researches, but it was a strength of the study. This proved that our PD model made accurate prediction even in very different study conditions and implied it could be useful in clinical practice. The comments are replied in an item-wise manner below. The submission system does not allow re-upload of the revised Appendix and Manuscript at this stage. We have the revisions written out and highlighted in the replies. Appendix is attached to the end due to its large size from major revision.

  1. The manuscript describes a novel approach for validation of the previously proposed by White et al. (Curr Drug Metab., 2003) pharmacodynamic model for three drugs given simultaneously to patients undergoing gastrointestinal endoscopies. The data for analysis were taken from literature. The study results may be interesting for clinicians, however, the manuscript is very difficult to follow and it is not possible to repeat the procedures used due to the lack of detailed information on the methodology.

Reply: Thank you. Here we present a new process for validation, hence the details described focused on how it is validated. The modeling details has been described by White et al and us in earlier studies (Curr Drug Metab 2003; 4(5): 399-409, Br J Anaesth 2018; 120(6): 1209-1218). A appendix file briefly described the PKPD models. The parameters and equations we used in this article are identical to published PKPD literatures. We have revised the appendix file greatly to enhance model descriptions.

(Revisions are too bulky to be placed here)

  1. First of all, the model used was not described in detail. It is not clear how the pharmacokinetic or pharmacodynamic interactions were incorporated in the model.

Reply: Thank you. Pharmacokinetics (PK) and pharmacodynamics (PD) must both exist to describe drug effects. PK describes how drugs are distributed but do not comment on any drug effect. PD describes the drug effects under the drug concentrations governed by PK. They describe drug behaviors in sequential order.

PK was calculated by a commercialized program that contained identical models used on a daily basis in clinical practice. The NLMAZ PD model underwent a comprehensive development process (Br J Anaesth 2018; 120(6): 1209-1218), using the same PK calculation techniques as in this study. The PKPD model details are not the main focus of the study and thus a short description was provided in the appendix material. We have revised the appendix file to clarify the relationship of PK and PD, now it reads,

Appendix file, Beginning

The study involved series of pharmacokinetic and pharmacodynamic simulations. The general approach is to calculate the pharmacokinetic effect-site drug concentrations (Ces) and use the Ces in the pharmacodynamic model to calculate drug effects. The pharmacokinetic and pharmacodynamic models presented here are published models. Model selection and model fit are not performed as a part of the study. The models are described below.

(Remaining revisions are too bulky to be placed here)

  1. In Appendix materials can be found thatThere are a total of 23 parameters for the MOAA/S < 2 model used for the simulation”. These parameters were not listed and their values used for simulations were not provided. From eq. 2 the number of parameters seems much lower.

Reply: Thank you. It’s true we did not list the full form of the model in the manuscript. It would greatly increase the article size. Each parameter required detailed descriptions, but the individual definitions were not used in the article. We listed the ones (C50) that were relevant to the main body. The complete model equations and the parameters were added to the appendix file.

(Revisions are too bulky to be placed here)

  1. In the sentence: “C50 resides in U50and stands for the concentration of a drug to reach half maximal effect” it is not clear whether C50 is plasma or effect site concentration.

C50mid=51.215, C50alf=627.933 and C50prop=2.581” – units are missing. They were called primary parameters, whereas they are not present in eq. 2 as model parameters.

Reply: Thank you for pointing them out. We have revised the appendix materials to enhance clarity and specificied they were effect-site concentrations. Now it reads,

Manuscript, 2.3 Pharmacologic models

Drug concentrations effects are calculated with the models. Opioids are converted to alfentanil equivalents (Appendix S1). All drug concentrations are expressed as Ce.

Appendix file, Beginning,

…The general approach is to calculate the pharmacokinetic effect-site drug concentrations (Ces) and use the Ces in the pharmacodynamic model to calculate drug effects. …

Appendix file, The Pharmacodynamic Model,

The primary parameters are C50mid=51.215 ng/mL, C50alf=627.933 ng/mL and C50prop=2.581 mcg/mL, which represent the C50 of midazolam, alfentanil and propofol respectively. The complete parameters are listed in Appendix Table 1.

  1. Line 113 – “Pharmacokinetic models are selected based on clinical relevance.” Usually they are selected based on the goodness of fit criteria. The PK data and models should be presented and model selection justified based on the standard criteria.

Reply: Thank you. The article is primarily a pharmacodynamic study. The PK and PD models used here were already developed and validated in earlier studies, using the goodness-of-fit criterion suggested by the reviewer. PK data are not available because we did not perform the PK goodness-of-fit. There are many PK models available for a single drug but clinically only 2 or 3 models are used (clinical relevance) because of superior accuracy in different scenarios. We chose the models that are used clinically, rather than arbitrary. We have revised the sentence to improve clarity, and now it reads,

Manuscript, 2.3 Pharmacologic models,

Pharmacokinetic models are selected based on availability and clinical relevance practice.

  1. Line 114 – “Drug concentrations effects are calculated with the models”. How were effects calculated with PK models?

Reply: Thank you. The program TIVAtrainer cited in the study is a commercialized simulation program that calculates both plasma (Cps) and effect-site (Ces) drug concentrations. It’s a powerful tool and matches the calculations made by clinical infusion smart pumps used in daily practice. The calculated Ces are input into the PD model and gives a drug effect probability for that given Ce set. PK model per se does not give drug effects. We apologize for the lack of description and made changes in the appendix file. Now it reads,

Appendix, Beginning,

The general approach is to calculate the pharmacokinetic effect-site drug concentrations (Ces) and use the Ces in the pharmacodynamic model to calculate drug effects.

Appendix, Pharmacokinetic simulation,

Pharmacokinetic profiles Effect-site concentrations (Ces) are calculated with TIVA trainer (Version 9.1, Build 6, EuroSIVA).

Appendix, The pharmacodynamic model, below equation 4,

The variables Cm, Ca and Cp refer to the calculated Ce of midazolam, alfentanil, and propofol respectively. For consistency throughout the article, the subscripts m, a, p will refer as midazolam, alfentanil and propofol respectively.

  1. The Authors have not presented how the study results may be used in clinical practice.

Reply: Thank you for the important comment. We presented a new method of model validation for pharmacodynamic models. Traditionally it takes a long time to validate a PD model starting from acquiring permission from the hospital administration, followed by patient recruitment. In the end the study group is still bound by hospital practice standards. The model may not be as accurate if used outside the hospital or under the influence of different endoscopists or anesthesiologists.

This new method overcomes the concerns by inclusion of a variety of endoscopy conditions, ranging from dosing behavior, procedural (gastroscope, colonoscopy or ERCP) and human factors (different patient demographics, endoscopists and anesthesiologists) from published literatures.

The study implies that the need for repeated volunteer or patient recruitments can be reduced for validation. The process is greatly shortened. The results of the study bring the model to clinical use at a faster pace. The model has can be used to guide clinical sedation or serve as simulation for anesthesiology training. We have revised the manuscript to highlight its impact, now it reads,

Manuscript, conclusion,

It potentially simplified pharmacodynamic research and reduce patient or volunteer exposure to anesthetic drugs. The model development process is greatly shortened and made the models available for clinical use at a quicker pace. We believe the models would bring meaningful clinical guidance and educational applications in sedation management.

Appendix S1. Pharmacokinetic and Pharmacodynamic Models

The study involved series of pharmacokinetic and pharmacodynamic simulations. The general approach is to calculate the pharmacokinetic effect-site drug concentrations (Ces) and use the Ces in the pharmacodynamic model to calculate drug effects. The pharmacokinetic and pharmacodynamic models presented here are published models. Model selection and model fit are not performed as a part of the study. The models are described below.

Pharmacokinetic simulation

Effect-site concentrations (Ces) are calculated with TIVA trainer (Version 9.1, Build 6, EuroSIVA). The simulation is based on a female who is 60-year-old, 170 cm and 65kg.1, 2 The Maitre model3 was used for alfentanil, Zomorodi model4 for midazolam, Shafer model5 for fentanyl and Schnider model6 for propofol. Ces are calculated for 60 minutes at 5-second intervals for better temporal resolution. Opioids are converted to alfentanil equivalents based on the potency ratio fentanyl:alfentanil:remifentanil = 1:0.0625:1.27. This process is necessary to compare different opioids and is a common technique used in anesthesia pharmacodynamic studies8, 9.

The Pharmacodynamic model

The Non-linear Mixed Effect with Zero amounts ( NLMAZ ) three-drug response surface model is used as our simulation model.10 The general form of the model follows the sigmoid-Emax curve (Equation 2):

[2]

E is the effect, defined as the probability of LOR. Modified OAA/S (MOAA/S) < 2 is used to represent LOR. Emax is the maximal drug effect possible and E0 is the baseline probability when no drugs are present. U50 and is the drug concentrations required for the drug U to take 50% maximal effect, that is, to achieve 50% chance of LOR. Parameter m is the slope factor that determines the steepness of the LOR change10. U can be interpreted as a new drug and is the sum of the normalized potency of midazolam, alfentanil and propofol (equation 3 and 4):

[3]

[4]

The variables Cm, Ca and Cp refer to the calculated Ce of midazolam, alfentanil, and propofol respectively. For consistency throughout the article, the subscripts m, a, p will refer as midazolam, alfentanil and propofol respectively. C50 is defined as the concentration of drug required to provide half maximal effect. For the model to scale correctly, we have to define:

[5]

[6]

Where x, y and z are the drug fractions of midazolam, alfentanil and propofol. The unknown parameters (P) are defined using the full cubic form of the canonical polynomial, as in equation 7:

[7]

This can be expanded into:

[8]

The Greek letter constants are referred to as the vector constants. Equation 8 is a generalized form of the parameters n and U50, where they substitute P and each has their designated vector constants:

[9]

[10]

There are a total of 23 parameters for the MOAA/S < 2 model used for the simulation. The primary parameters are C50mid=51.2 ng/mL, C50alf=627.9 ng/mL and C50prop=2.58 mcg/mL10, which represent the C50 of midazolam, alfentanil and propofol respectively. The complete parameters are listed in Appendix Table 1.

Appendix Table 1. Model parameters.

Parameter

Value

Parameter

Value

C50,alf

51.2

δD123

-0.31

C50,mid

627.9

αm1

5.51

C50,prop

2.58

αm2

9.4

αD1

-0.21

αm3

9.3

αD2

-0.26

βm12

2.1

αD3

-0.23

βm13

-0.2

βD12

-0.99

βm23

9.5

βD13

-0.14

γm12

-0.0075

βD23

-0.3

γm13

-0.0078

γD12

-0.07

γm23

-0.0098

γD13

-0.03

δm123

-15.41

γD23

0.08

Greek letters are the vector constants of the pharmacodynamic model.

References:

  1. VanNatta ME, Rex DK. Propofol alone titrated to deep sedation versus propofol in combination with opioids and/or benzodiazepines and titrated to moderate sedation for colonoscopy. Am J Gastroenterol 2006;101:2209-17.
  2. Levitzky BE, Lopez R, Dumot JA, Vargo JJ. Moderate sedation for elective upper endoscopy with balanced propofol versus fentanyl and midazolam alone: a randomized clinical trial. Endoscopy 2012;44:13-20.
  3. Maitre PO, Vozeh S, Heykants J, Thomson DA, Stanski DR. Population pharmacokinetics of alfentanil: the average dose-plasma concentration relationship and interindividual variability in patients. Anesthesiology 1987;66:3-12.
  4. Zomorodi K, Donner A, Somma J, et al. Population pharmacokinetics of midazolam administered by target controlled infusion for sedation following coronary artery bypass grafting. Anesthesiology 1998;89:1418-29.
  5. Shafer SL, Varvel JR, Aziz N, Scott JC. Pharmacokinetics of fentanyl administered by computer-controlled infusion pump. Anesthesiology 1990;73:1091-102.
  6. Schnider TW, Minto CF, Gambus PL, et al. The influence of method of administration and covariates on the pharmacokinetics of propofol in adult volunteers. Anesthesiology 1998;88:1170-82.
  7. Liou JY, Tsou MY, Ting CK. Response surface models in the field of anesthesia: A crash course. Acta Anaesthesiol Taiwan 2015;53:139-45.
  8. Vereecke HE, Proost JH, Heyse B, et al. Interaction between nitrous oxide, sevoflurane, and opioids: a response surface approach. Anesthesiology 2013;118:894-902.
  9. Vuyk J, Mertens MJ, Olofsen E, Burm AG, Bovill JG. Propofol anesthesia and rational opioid selection: determination of optimal EC50-EC95 propofol-opioid concentrations that assure adequate anesthesia and a rapid return of consciousness. Anesthesiology 1997;87:1549-62.
  10. Liou JY, Ting CK, Teng WN, Mandell MS, Tsou MY. Adaptation of non-linear mixed amount with zero amount response surface model for analysis of concentration-dependent synergism and safety with midazolam, alfentanil, and propofol sedation. Br J Anaesth 2018;120:1209-18.

Reviewer 2 Report

Line 24: It is mentioned that 9 published clinical studies were included. However line 137 mentioned that only 6 eligible full-articles were included, with a total of 9 regimen. So is the earlier sentence referring to 9 regimens or the 6 published articles included?

Line 25: Specify the names of the opioids since other drug names (midazolam and propofol) are mentioned.

Line 82: incomplete sentence. Also check for grammar errors across the text, there were a number of improperly used sentences.

Line 96: Provide complete definition for the abbreviation MOAA/S before using it in the text.

Line 103: Provide complete definition for the abbreviation Ce before using it in the text

Line 114: Provide further details for the reader to better understand the reason for why opioids are converted to alfentanil equivalent

Line 230: In this paragraph, some inconsistencies are highlighted with dose amounts and extent of sedation. It will help if the demographics for population in the studies included in this  manuscript are provided in a tabular format and described in results. Perhaps, this can explain the reason for the differences observed

Line 112: Despite the fact that these models were previously developed and validated, it will help to describe it in more detail to the reader before presenting the results obtained from using it. More so, the model parameter values used in the simulation needs to be provided.

Did the authors consider any potential covariates that may affect the prediction outcomes?

Author Response

(The replies are best read by word processors)

Reviewer 2

We thank the reviewer for pointing out important issues and provided constructive comments. The comments are replied in an item-wise manner below. The submission system does not allow re-upload of the revised Appendix and Manuscript at this stage. We have the revisions written out and highlighted in the replies. Appendix is attached to the end due to its large size from major revision.

  1. Line 24: It is mentioned that 9 published clinical studies were included. However line 137 mentioned that only 6 eligible full-articles were included, with a total of 9 regimen. So is the earlier sentence referring to 9 regimens or the 6 published articles included?

Reply: Thank you. We ended up with 6 clinical studies after the described search method. Three of which has 2 regimens available for simulation, giving us a total of 9 regimens to work with. We apologize for the typo in the abstract. It is revised and now reads,

Manuscript, Abstract,

…we propose a new method of model validation for recovery and intraprocedural sedation adequacy with a three-drug pharmacodynamic model using nine six published clinical studies that contain midazolam, opioid and propofol. …

  1. Line 25: Specify the names of the opioids since other drug names (midazolam and propofol) are mentioned.

Reply: Thank you. There were alfentanil, fentanyl and remifentanil of the opioid family in the article. We used the more general term ‘opioid’. It is a deliberate choice to acknowledge the applicability for other opioids as well. This is usually done through conversion of the opioids according to their potency ratios. The approach and phrasing is common in anesthesiology pharmacodynamic studies (Mathematics 2022; 10(10): 1651, Anesthesiology 2006; 105(2): 267-278, Anesthesiology 2012; 117(2): 252-262, Anesthesiology 2013; 118(4): 894-902). The appendix file has been revised to add clarity, and now reads,

Appendix file, Pharmacokinetic simulation,

…1:0.0625:1.2. This process is necessary to compare different opioids and is a common technique used in anesthesia pharmacodynamic studies.

  1. Line 82: incomplete sentence. Also check for grammar errors across the text, there were a number of improperly used sentences.

Reply: Thank you. The manuscript received English editing service prior to submission. We will check again with the service for the grammars.

  1. Line 96: Provide complete definition for the abbreviation MOAA/S before using it in the text.

Reply: Thank you for pointing out. We have revised the manuscript and table description, and now they read,

Manuscript, 2.2 simulation setup,

The 5% probability of MOAA/S (Modified OAA/S, Table 1) < 2 is used as a cutoff to identify model predicted recovery from sedation for the simulation.

Table 1, table description

The original OAA/S does not have score 0.

  1. Line 103: Provide complete definition for the abbreviation Ce before using it in the text

Reply: Thank you. We apologize for the error. It is revised and now reads,

Manuscript, 2.2 Simulation setup

Fentanyl reaches peak effect-site concentration (Ce) at 3.6 minutes and more…

  1. Line 114: Provide further details for the reader to better understand the reason for why opioids are converted to alfentanil equivalent

Reply: Thank you. The reason is the same as the reply in item 2. We used the general term ‘opioids’ to imply the model’s applicability most opioids, provided the conversion ratio is known. In order to compare different opioids, we chose alfentanil as the base and all opioids converted to alfentanil equivalents. Other opioids can be used as base and the results would be the same. We added the reasoning in the Appendix file, and now reads,

Appendix file, Pharmacokinetic simulation,

…1:0.0625:1.2. This process is necessary to compare different opioids and is a common technique used in anesthesia pharmacodynamic studies.

  1. Line 230: In this paragraph, some inconsistencies are highlighted with dose amounts and extent of sedation. It will help if the demographics for population in the studies included in this manuscript are provided in a tabular format and described in results. Perhaps, this can explain the reason for the differences observed

Reply: Thank you, it is an important aspect to consider. The purpose of the paragraph was to present the fact that different sedation responses can occur with similar total drug doses. We have considered demographic factors as possible factors during our initial analyses but discovered that they do not provide sufficient explanation. For example, the mean age of the two regimens mentioned in the paragraph (Line 230-238) were 55.4 and 61.6, with slight female predominance in both studies. Other factors are insufficient to draw conclusions. Also, there are great variations in the reporting of demographic across the six studied. Some do not report weights, heights or ASA PS (American Society of Anesthesiologist Physical Status) score. As a result, the demographics was unable to explain the sedation response differences. Revision to the manuscript was made and now reads,

Manuscript, Discussion, Paragraph 7 (Line 238)

Therefore, we reasoned that the study reported sedation scores that did not agree with our predictions might have risen from interrater variations. Other demographic variables were considered but they were unable to explain the difference between observation and prediction.

  1. Line 112: Despite the fact that these models were previously developed and validated, it will help to describe it in more detail to the reader before presenting the results obtained from using it. More so, the model parameter values used in the simulation needs to be provided.

Reply: Thank you. A large section of the modeling details is added to the Appendix, including the parameter descriptions and their values.

(Apologies, the revisions are too bulky to be placed here)

  1. Did the authors consider any potential covariates that may affect the prediction outcomes?

Reply: The predictions made by the model is minimally affected by demographic variations. There are age, weight or gender differences in pharmacokinetics, but the calculated effect-site concentrations (Ce) did not differ much.

Age is a very important covariate to consider in pharmacodynamics. Younger patients tend to require far more drugs than older patients to achieve the same effect. The model was initially built with middle-aged patients, which was consistent with the studies, age-wise. This makes the age covariate less influential on the prediction differences. Other clinical covariates such as endoscopist techniques may play a role but they are not reported in the studies and no conclusions can be drawn from the available information. We focused on the inter-observer differences in sedation evaluation because they are reported for comparison. Revision to the manuscript was added to address the issue and now reads,

Manuscript, Discussion, Paragraph 7 (Line 238)

Therefore, we reasoned that the study reported sedation scores that did not agree with our predictions might have risen from interrater variations. Other demographic variables were considered but they were unable to explain the difference between observation and prediction.

Appendix S1. Pharmacokinetic and Pharmacodynamic Models

The study involved series of pharmacokinetic and pharmacodynamic simulations. The general approach is to calculate the pharmacokinetic effect-site drug concentrations (Ces) and use the Ces in the pharmacodynamic model to calculate drug effects. The pharmacokinetic and pharmacodynamic models presented here are published models. Model selection and model fit are not performed as a part of the study. The models are described below.

Pharmacokinetic simulation

Effect-site concentrations (Ces) are calculated with TIVA trainer (Version 9.1, Build 6, EuroSIVA). The simulation is based on a female who is 60-year-old, 170 cm and 65kg.1, 2 The Maitre model3 was used for alfentanil, Zomorodi model4 for midazolam, Shafer model5 for fentanyl and Schnider model6 for propofol. Ces are calculated for 60 minutes at 5-second intervals for better temporal resolution. Opioids are converted to alfentanil equivalents based on the potency ratio fentanyl:alfentanil:remifentanil = 1:0.0625:1.27. This process is necessary to compare different opioids and is a common technique used in anesthesia pharmacodynamic studies8, 9.

The Pharmacodynamic model

The Non-linear Mixed Effect with Zero amounts ( NLMAZ ) three-drug response surface model is used as our simulation model.10 The general form of the model follows the sigmoid-Emax curve (Equation 2):

[2]

E is the effect, defined as the probability of LOR. Modified OAA/S (MOAA/S) < 2 is used to represent LOR. Emax is the maximal drug effect possible and E0 is the baseline probability when no drugs are present. U50 and is the drug concentrations required for the drug U to take 50% maximal effect, that is, to achieve 50% chance of LOR. Parameter m is the slope factor that determines the steepness of the LOR change10. U can be interpreted as a new drug and is the sum of the normalized potency of midazolam, alfentanil and propofol (equation 3 and 4):

[3]

[4]

The variables Cm, Ca and Cp refer to the calculated Ce of midazolam, alfentanil, and propofol respectively. For consistency throughout the article, the subscripts m, a, p will refer as midazolam, alfentanil and propofol respectively. C50 is defined as the concentration of drug required to provide half maximal effect. For the model to scale correctly, we have to define:

[5]

[6]

Where x, y and z are the drug fractions of midazolam, alfentanil and propofol. The unknown parameters (P) are defined using the full cubic form of the canonical polynomial, as in equation 7:

[7]

This can be expanded into:

[8]

The Greek letter constants are referred to as the vector constants. Equation 8 is a generalized form of the parameters n and U50, where they substitute P and each has their designated vector constants:

[9]

[10]

There are a total of 23 parameters for the MOAA/S < 2 model used for the simulation. The primary parameters are C50mid=51.2 ng/mL, C50alf=627.9 ng/mL and C50prop=2.58 mcg/mL10, which represent the C50 of midazolam, alfentanil and propofol respectively. The complete parameters are listed in Appendix Table 1.

Appendix Table 1. Model parameters.

Parameter

Value

Parameter

Value

C50,alf

51.2

δD123

-0.31

C50,mid

627.9

αm1

5.51

C50,prop

2.58

αm2

9.4

αD1

-0.21

αm3

9.3

αD2

-0.26

βm12

2.1

αD3

-0.23

βm13

-0.2

βD12

-0.99

βm23

9.5

βD13

-0.14

γm12

-0.0075

βD23

-0.3

γm13

-0.0078

γD12

-0.07

γm23

-0.0098

γD13

-0.03

δm123

-15.41

γD23

0.08

Greek letters are the vector constants of the pharmacodynamic model.

References:

  1. VanNatta ME, Rex DK. Propofol alone titrated to deep sedation versus propofol in combination with opioids and/or benzodiazepines and titrated to moderate sedation for colonoscopy. Am J Gastroenterol 2006;101:2209-17.
  2. Levitzky BE, Lopez R, Dumot JA, Vargo JJ. Moderate sedation for elective upper endoscopy with balanced propofol versus fentanyl and midazolam alone: a randomized clinical trial. Endoscopy 2012;44:13-20.
  3. Maitre PO, Vozeh S, Heykants J, Thomson DA, Stanski DR. Population pharmacokinetics of alfentanil: the average dose-plasma concentration relationship and interindividual variability in patients. Anesthesiology 1987;66:3-12.
  4. Zomorodi K, Donner A, Somma J, et al. Population pharmacokinetics of midazolam administered by target controlled infusion for sedation following coronary artery bypass grafting. Anesthesiology 1998;89:1418-29.
  5. Shafer SL, Varvel JR, Aziz N, Scott JC. Pharmacokinetics of fentanyl administered by computer-controlled infusion pump. Anesthesiology 1990;73:1091-102.
  6. Schnider TW, Minto CF, Gambus PL, et al. The influence of method of administration and covariates on the pharmacokinetics of propofol in adult volunteers. Anesthesiology 1998;88:1170-82.
  7. Liou JY, Tsou MY, Ting CK. Response surface models in the field of anesthesia: A crash course. Acta Anaesthesiol Taiwan 2015;53:139-45.
  8. Vereecke HE, Proost JH, Heyse B, et al. Interaction between nitrous oxide, sevoflurane, and opioids: a response surface approach. Anesthesiology 2013;118:894-902.
  9. Vuyk J, Mertens MJ, Olofsen E, Burm AG, Bovill JG. Propofol anesthesia and rational opioid selection: determination of optimal EC50-EC95 propofol-opioid concentrations that assure adequate anesthesia and a rapid return of consciousness. Anesthesiology 1997;87:1549-62.
  10. Liou JY, Ting CK, Teng WN, Mandell MS, Tsou MY. Adaptation of non-linear mixed amount with zero amount response surface model for analysis of concentration-dependent synergism and safety with midazolam, alfentanil, and propofol sedation. Br J Anaesth 2018;120:1209-18.

Round 2

Reviewer 1 Report

The authors have satisfactorily responded to all my questions and comments. In particular, the description of the model used is much more improved in the revised manuscript and the supplementary materials.